# Synthesizing perspectives: Crafting an Interdisciplinary view of social media's impact on young people's mental health

John Maltby[1]*, Thooba Rayes[2], Antara Nage[1], Sulaimaan Sharif[2], Maryama Omar[2], Sanjiv Nichani[3]

**1** School of Psychology and Vision Sciences, University of Leicester, Leicester, Leicestershire, United Kingdom, **2** School of Medicine, University of Leicester, Leicester, Leicestershire, United Kingdom, **3** Leicester Children's Hospital, University Hospitals of Leicester NHS Trust, Leicester, Leicestershire, United Kingdom

* jm148@leicester.ac.uk

**Data Availability Statement:** The data is available at the University of Leicester Figshare Data Repository: https://figshare.com/s/582c78d2302fa049bf3a.

## Abstract

This study explores the intricate relationship between social media usage and the mental health of young individuals by leveraging the insights of 492 UK school headteachers. It adopts a novel multidisciplinary approach, integrating perspectives from psychology, sociology, education studies, political science, philosophy, media studies, linguistics, social work, anthropology, and health sciences. The application of thematic analysis, powered by ChatGPT-4, identifies a predominantly negative perspective on the impact of social media on young people, focusing on key themes across various disciplines, including mental health, identity formation, social interaction and comparison, bullying, digital literacy, and governance policies. These findings culminated in the development of the five-factor Comprehensive Digital Influence Model, suggesting five key themes (Self-Identity and Perception Formation, Social Interaction Skills and Peer Communication, Mental and Emotional Well-Being, Digital Literacy, Critical Thinking, and Information Perception, and Governance, Policy, and Cultural Influence in Digital Spaces) to focus the impacts of social media on young peoples' mental health across primary and secondary educational stages. This study not only advances academic discourse across multiple disciplines but also provides practical insights for educators, policymakers, and mental health professionals, seeking to navigate the challenges and opportunities presented by social media in the digital era.

## Introduction

In today's digital landscape, the widespread use of social media among young people sparks waves of concern, reminiscent of historical anxieties about the influence of emerging media forms like rock music, television, and video games [1–4]. Unlike its predecessors, social media uniquely combines continuous connectivity, interactivity, and personalisation, thus creating a rich tapestry of engagement that profoundly affects the mental health of its users [5–7]. Studies reveal a complex relationship wherein social media can be a source of support and connection,

**Funding:** The author(s) received no specific funding for this work.

**Competing interests:** The authors have declared that no competing interests exist.

yet it also harbours the potential to exacerbate issues like anxiety, depression, and sleep disturbances [7–10]. This nuanced impact underscores the need for a multidisciplinary approach to fully understand and address the mental health challenges posed by social media to young people.

The intricate relationship between social media use and mental health challenges, particularly in young people, has been the focus of extensive academic scrutiny across various disciplines, providing theoretical contexts with which to consider the dynamics between social media and mental health among young people. For example, *Psychology* offers critical insights into the individual-level mental health effects of social media, examining how online interactions are related to mental health negatively (lower self-esteem, lower mood, and anxiety) and positively (social connectedness and meaning) [11, 12]. *Sociology* provides a broader lens, exploring the social implications of social media beyond the individual, thus allowing for an understanding of the collective phenomena associated with social media use, including the formation of online communities, and the spread of social norms. As such sociological research emphasises how social media not only shapes personal identities but also influences wider social networks and societal structures [13, 14]. *Education Studies* emphasize the promotion of adolescents' mental health through experiential and interactive activities, positive education frameworks, and fostering digital responsibility. They highlight the crucial role of teachers and student engagement in integrating these practices into everyday educational settings to support overall well-being [15–17]. *Political science* explores how social media platforms function not only as arenas for personal interaction and self-expression but also as significant spaces for political socialisation, civic engagement, and mobilisation among young people [18]. *Philosophical perspectives* explore the ethical, moral, and existential questions surrounding social media use, probing into the ethical considerations of online behaviour, privacy, and the digital self, offering a critical perspective on the influence of social media on young people's lives [19]. *Media studies* explore the relationship between social media use and young people's mental health introducing and utilising concepts like cyberbullying, digital literacy, social media addiction, and information overload to consider social media's effects on society and individual well-being and the importance of navigational skills in digital environments [20, 21]. *Linguistics* sheds light on how social media transforms language use and communication patterns; revealing the complexities of digital communication, including the role of emojis, memes, and online slang, in shaping interactions and social connections. Findings suggest that the analysis of the digital linguistic exchanges on social platforms, provide insights into the evolving nature of language, discourse, and communication in the digital age [22]. *Social work* perspectives emphasise practical approaches to supporting young people facing mental health challenges exacerbated by social media, by discussing social and policy interventions aimed at mitigating the negative impacts of social media, thus highlighting the importance of support mechanisms, educational programs, and policy frameworks designed to protect and empower young people [23]. *Anthropological* perspectives can assist in examining the cultural and societal dimensions of social media; exploring how it reshapes human interactions, community bonds, and cultural identities. For example, research suggests how social media impacts young peoples' perceptions and societal roles, offering the study of cultural practices and narratives that emerge within digital spaces [24, 25]. Finally, *health sciences* perspectives explore social media influences on promoting risky behaviors, but also social support and information access information can mitigate risks and enhance positive impacts [26–28].

The complex relationship between social media use and young persons' mental health necessitates a multidisciplinary approach to fully comprehend and effectively address its multifaceted nature. The lack of integration among various academic disciplines often results in fragmented insights, with disciplines such as sociology, psychology, education, media studies,

health sciences, political science, social work, anthropology, philosophy, and linguistics operating in silos, leading to a partial and segmented understanding. This division hinders the development of comprehensive interventions and policy strategies. To overcome these limitations, an integrated, interdisciplinary perspective is crucial–that would aim for a more holistic account of the topic, recognizing the complexity, and present a more succinct account appropriate for policy and educational intervention. First, combining different disciplines could provide a more holistic understanding of impact. For example, whereas a psychology-focused study might exclusively examine the direct effects of social media on individual mental health symptoms like anxiety and depression [29], an interdisciplinary approach by integrating psychology, sociology, media studies, and education offers a broader understanding that encompasses not only these direct effects but also the underlying social dynamics, educational impacts, and media interactions [30–32]. This holistic view reveals how these areas intersect and influence one another, leading to more nuanced findings. Second, using different disciplines might acknowledge where levels of complexity exist. For example, while an anthropology-focused study might limit itself to exploring cultural impacts and identity formation [24], combining anthropology with psychology and sociology allows for an exploration of how cultural impacts influence mental health and social behavior in a feedback loop [33]. Such an interdisciplinary approach acknowledges complex relationships, for instance, how cultural norms around social media usage can exacerbate or mitigate mental health issues. Third, work then that aims to support educational/policy recommendations also benefit from this interdisciplinary synergy. Instead of a narrow focus from political science on governance and regulation [34], merging insights from political science, social work, health sciences, and media studies allows for the formulation of more nuanced policy recommendations [35]. These can address not just the regulation of digital content but also the use of social media to promote mental health resources, community engagement strategies, and educational reforms [8], thereby enriching societal and cultural implications. In summary, the interdisciplinary confluence is imperative not only for addressing the symptoms of social media's influence but also for achieving a cohesive and deeper understanding of how social media shapes both individual identities and broader societal dynamics. This will be accomplished by developing a holistic view, recognizing the complexity of these interactions, and providing succinct, actionable policy and educational recommendations.

This study aims to delve into the effects of social media on young people's mental health by merging insights from various academic fields. By synthesizing knowledge from disciplines such as psychology, sociology, education, and media studies, we form a holistic understanding of how social media influences the well-being of young individuals. Specifically, this study seeks to answer the research question: 'How do interdisciplinary perspectives enhance our understanding of the impact of social media on the mental health of young people?' This question aims to clarify the scope of our research and guide the investigation into the complex dynamics of social media's influence across different disciplinary lenses.

## Method

Ethics approval for the study was granted by the University of Leicester Psychology Ethics Board, with the reference number 42292-jm148-ls&visionsciences,schof. In alignment with ethical guidelines, informed consent was obtained from all participants. This consent was facilitated through an online survey, where respondents indicated their consent by affirmatively selecting the consent option on the digital form confirming consent statements. This consent option confirmed that participants were fully informed and voluntarily agreed to participate in the study.

## Sample

The identification of school headteachers as a cohort was strategic to pursue a nuanced exploration of the multidisciplinary examination of impact of social media, on the mental health of young individuals. The inclusion of headteachers in this multidisciplinary study is key for three reasons. First, Headteachers provide a unique interdisciplinary perspective in the study of social media's impact on young people's mental health. Their broad understanding across various disciplines and factors relevant to children (e.g., educational theory through its application in practice, psychological aspects via counselling and supporting children, sociological elements through interactions with families and communities within schools, political theory by working with local and national policies, social work through engagement in themes of social care) helps in effectively addressing the complex nature of this issue. This multidisciplinary approach allows headteachers to bring practical and experiential knowledge from their roles in educational settings. Second, they offer a panoramic view of school dynamics, observing social interactions and community issues. This comprehensive overview aids in identifying patterns of social media use and its effects on student well-being, contributing to a real-world understanding of its implications for mental health. Third, Headteachers are pivotal in linking policy with mental health awareness among young people. Their experiences in tackling digital challenges and shaping school policies are instrumental in creating balanced strategies for social media use. They advocate for mental health resources, enhancing the learning environment's safety and supportiveness.

We surveyed 534 individuals from UK schools, reaching out to approximately 25,000 school email addresses directed to the attention of the headteacher. After excluding 41 respondents due to their roles as teachers, teaching assistants, IT staff, or age-related criteria, we analysed data from 492 participants. This group comprised 109 males, 379 females, and 4 individuals identifying as 'other', with an average age of 50.29 years (SD = 6.7). Respondents held roles in management, predominantly as Headteachers (442), Deputy Heads (50), or Heads of Pupil Welfare (11), representing a wide geographical spread across the UK, including England (416; most prominently, East of England, $n = 86$; South East England, $n = 83$; West Midlands, $n = 55$, North England $n = 53$; and Yorkshire and Humber, $n = 43$), Scotland (45), Wales (20), and Northern Ireland (9) (2 individuals did not supply this information), with diverse school sizes in both primary (n = 171) and secondary education sectors. One hundred and seventy-one respondents were in charge of a primary school, and 321 oversaw a secondary school. Based on the data provided by the UK Government [36], which states that the average size of primary schools in the UK is 280 pupils, and secondary schools is 980 pupils, our study received varied responses from teachers in schools of different sizes. Specifically, 126 responses were from teachers in primary schools smaller than the average size, while 45 responses were from teachers in primary schools larger than the average size. In the case of secondary schools, 255 responses originated from teachers in schools smaller than the average size, and 66 responses were from headteachers in secondary schools that exceeded the average size.

## Procedure

Respondents completed an online survey. The recruitment period for participants commenced in November 2023, and concluded in January 2024. Aside from question to establish the demographic data provided above, responses were primarily focused on assessing the perceived impact of social media on the mental health of children and young adults. The question posed to respondents was, "In your professional opinion, to what extent does social media use in children, young people and young adults influence their mental health". We used a singular question to capture their unique insight efficiently and effectively on social media's impact on

mental health and so to encourage engagement. The question was also careful to leverage their broad professional experiences and aimed to encourage reflective, in-depth responses, pertinent to developing targeted educational and mental health strategies.

## Data analysis

**General approach.** In the context of contemporary research, adopting a *qualitative*, *multidisciplinary approach* necessitates engagement with a methodological framework that is inherently broad and subject to ongoing development [35, 37, 38]. The absence of a singular, universally accepted methodology within this domain underscores the complexity and dynamism inherent in integrating diverse disciplinary perspectives. Such an approach is predicated on the premise that a multifaceted examination of research questions can yield a more nuanced and comprehensive understanding of the phenomena under study [35, 37, 38].

We propose a qualitative multidisciplinary analysis method for analyzing headteachers' perspectives on social media's influence on young people's mental health. This method employs thematic analysis across multiple disciplines, followed by the creation of superordinate themes. By integrating thematic analysis across various disciplines and subsequently synthesizing superordinate themes, this approach not only harnesses the power of interdisciplinary insights but also leverages the capabilities of ChatGPT-4 to elevate the process to new levels of precision and depth. ChatGPT-4 excels in a wide array of natural language processing tasks, demonstrating significant capabilities in text summarization, and performs well on natural and large language inference tasks, showing comparable performance to other state-of-the-art models in question-answering and summarization tasks [39–41]. This methodology stands out for its comparison to traditional qualitative analysis techniques, primarily through its employment of ChatGPT-4 as a tool for data interpretation. Comparable to content and assessment tasks completed by humans [42, 43], ChatGPT-4 potentially offers a consistent, unbiased processing capability, crucial for handling the subtle and often subjective nature of qualitative data. Consequently, when compared to a set of human researchers, ChatGPT-4's advanced algorithms enable it to dissect and interpret quotes within a multitude of academic contexts, ensuring that every piece of data is analyzed thoroughly and from multiple perspectives. It can also handle tasks or recognize objects, concepts, or activities that it has not seen during training. This zero-shot learning capability is particularly valuable because it enables the model to generalize from its existing knowledge without needing explicit examples of every possible scenario it might encounter. However, despite improvements [44], it still retains some level of political bias influenced by factors like the languages used in the system [45–47]. This could affect its neutrality in applications, presenting a caveat in the deployment of this model in multidisciplinary qualitative analysis where unbiased perspectives are critical. It was felt that the potential political bias of ChatGPT-4 is less concerning when analyzing headteachers' views on social media's impact on youth mental health. The focus on educational outcomes, reliance on professional insights, and broad agreement on the importance of addressing youth mental health issues help minimize the influence of any bias.

Moreover, the significance of this process extends beyond its innovative use of artificial intelligence. By drawing on insights from multiple disciplines, the methodology reveals the multidisciplinary nature of qualitative analysis that often remains hidden. Quotes are not just viewed through a single disciplinary lens but are understood to have multifaceted meanings that apply differently across various fields. This aspect of the analysis is crucial, as it highlights the overlapping and interconnected nature of themes, which might otherwise be overlooked in a more siloed approach. It underscores the importance of recognising that different academic disciplines can provide complementary insights, enriching our overall understanding of the

topic at hand. The synthesis of these insights into superordinate themes will capture the complexity of how social media influences young peoples' mental health but also facilitates the emergence of a meta-theory that consolidates these diverse perspectives into a unified understanding. This consolidation is paramount as it addresses a critical gap in traditional thematic analysis—the integration of multidisciplinary insights into a coherent narrative that reflects an opportunity for multi- and inter-disciplinary use. Therefore, generally, the analysis took three elements (which we describe in detail in the next section):

1. **Data Segregation**: We categorised responses based on the educational stage—primary and secondary schools—to align with the developmental stages of their student populations.

2. **Multidisciplinary Analysis**: Utilising ChatGPT-4, we conducted a ten-fold analysis per educational level, applying theories and terminologies from Sociology, Psychology, Education Studies, Political Science, Philosophy, Media Studies, Linguistics, Social Work, Anthropology, and Health Sciences.

3. **Thematic Analysis Process**:

   - Stage 1: ChatGPT-4 identified initial themes for each discipline from the responses, articulating definitions and providing illustrative quotes for a nuanced understanding.

   - Stage 2: We synthesised these initial themes into superordinate themes across disciplines, developing overarching theories that reflect the multifaceted impact of social media on mental health.

**Detailed approach.** *Data segregation*. This analysis was divided into two main segments based on the educational level: primary (maximum age of pupils being 11 or below) ($n$ = 171) and secondary school headteachers ($n$ = 321) (overseeing pupils above 11 but below 18 years). The responses from each segment were analysed separately to account for the differing developmental stages and social media impacts at these educational levels.

*Multidisciplinary analysis using ChatGPT-4*. Each set of data (primary and secondary) underwent a comprehensive analysis using ChatGPT-4, repeated ten times for each 10 academic disciplines. ChatGPT-4, an advanced AI language model, was employed to analyse the comments utilising theories and terminologies consistent with each discipline. Prior to each analysis, a specific discipline was selected from a list of ten disciplines and subdisciplines. These were Psychology, Sociology, Education Studies, Political Science, Philosophy, Media Studies, Linguistics, Social Work, Anthropology, and Health Sciences. The reason for including these 10 areas was as follows. **Psychology**: Crucial for understanding the cognitive, emotional, and behavioural impacts of social media on the mental health of children and adolescents [48–50]. **Sociology**: Provides insights into how social media shapes and is influenced by social structures, norms, and interactions, impacting the mental health of young individuals [13, 51]. **Education Studies**: Offers perspectives on the role of social media in educational environments and its influence on learning, development, and mental health of students [52]. **Political Science**: Examines the impact of social media on young people's political socialisation, civic engagement, and its broader implications for their mental well-being [53, 54]. **Philosophy**: Explores ethical, existential, and epistemological aspects of social media use, contributing to a deeper understanding of its impact on young people's mental health [55, 56]. **Media Studies**: Investigates the role of media content, technology, and communication patterns in shaping young people's perceptions, behaviours, and mental health [57]. **Linguistics**: Analyses how language use and communication in social media influence the mental health, identity, and social interactions of young people [58]. **Social Work**: Addresses the

practical implications of social media use on young people's mental health and provides frameworks for intervention and support [59, 60]. **Anthropology**: Offers insights into the cultural and societal dimensions of social media use and its impact on the mental health of children and young adults [61]. **Health Sciences**: Focuses on the physiological and psychological aspects of social media use, examining its effects on the overall mental and physical health of young people [62, 63].

*Analysis process*. The analysis process comprises two stages: (1) the identification of initial themes by discipline and (2) the creation of super-ordinate themes.

***Stage 1. Analysis processes used by ChatGPT-4 –Initial themes.*** ChatGPT-4 analysed the headteachers' comments, separately for primary and secondary headteachers, employing the language and theoretical frameworks of the selected discipline. The AI model was instructed (a step-by-step process is provided in S1 Table) to identify overarching themes from the headteacher statements for the 10 disciplines in turn to ensure that the analysis was deeply rooted in the specific academic disciplines. The process is best described in 3 steps in ChatGPT instructions. First, was **theme identification** for each of the ten disciplines using sub-disciplinary frameworks to describe each discipline (S2 Table). For each discipline ChatGPT-4 was asked to use identified key themes in the headteachers' responses, using the Sub-disciplinary Frameworks as a definition of the framework. This involved recognising patterns, sentiments, and central ideas relevant to the discipline's focus consistent with thematic analysis [64]. Braun and Clarke's thematic analysis approach enables multidisciplinary research through its rigorous, flexible approach, revealing nuanced insights and overlooked patterns, and providing a consistent framework to understand complex phenomena. Second, Chatgpt4 was asked to provide a **definition and explanation of each of the themes it identified,** by articulating a clear definition and provided a context-specific explanation, integrating relevant examples or scenarios as applicable. Third, was to provide **two illustrative quotes and justification** for the quote as illustrative of the disciplinary theme.

***Stage 2. Analysis processes used by ChatGPT-4: Superordinate themes.*** We then, separately for primary and then secondary school data, using ChatGPT, used structured thematic analysis to synthesise the themes from 10 distinct academic disciplines in Stage 1 into superordinate themes. The procedure (a step-by-step process is provided in S3 Table) involved several key steps. Key themes from each discipline were first identified and reviewed. These were then synthesised into superordinate themes, capturing commonalities and differences across disciplines. Each superordinate theme was comprehensively analysed, integrating diverse academic insights to form a multidimensional understanding. Overarching theories were developed for each theme, reflecting the complexity and interdisciplinary nature of the findings. Finally, these themes and theories were synthesised, presenting a cohesive, multifaceted perspective on the influence of social media on mental health.

## Results

Stage 1 of the analysis identified the initial themes for each of the ten academic disciplines, separately for primary and secondary headteachers. S4 Table provides a detailed breakdown of the initial themes identified for each discipline for *primary* school headteachers' views on social media's impact on young persons' mental health. From a psychological perspective, the themes cover self-perception, cyberbullying, social skill effects, and risks including inappropriate content. From a sociological perspective, themes emerge around self-image through unrealistic standards, peer interactions, including cyberbullying, and mental overload from constant connectivity. From an educational studies perspective, themes focus on digital literacy and teaching critical thinking skills, nurturing social-emotional skills, prioritising online

safety, and recognising social media's impact on self-identity. From a political science perspective, themes focus on parental and community engagement, policy impact on social behaviour, assess globalised cultural influence, and digital space governance for young persons' well-being. From a philosophy perspective, themes explore how social media shapes reality perception, influences behaviour and morality, impacts cognitive-emotional development, and plays a role in social dynamics and inclusion. From a media studies perspective, themes focus on how social media shapes self-identity, changes in social skills, media literacy, and fostering online communities and support networks for young people. From a linguistics perspective, the themes focus on impacts on mental health, distortions in self-perception and body image, fuelling of cyberbullying and peer pressure, worsens anxiety, fostering an 'always-on' culture, and disrupts real-world interactions. From a social work perspective, themes focus on affecting self-esteem, body image, fuelling online bullying and peer pressure, impacting mental health, the addictive nature, disrupting social skills, and altering family dynamics and communication skills. From an anthropology perspective, the themes focus on moulding cultural norms and expectations around conformity and appearance, and the paradox of both social connection and isolation. From a Health Science perspective, the themes focus on impacts on mental health and social comparison and self-esteem, fuelling cyber-bullying, influences on physical health, and parental influence and modelling.

S5 Table provides a detailed breakdown of the initial themes identified for each discipline for *secondary* school headteachers in views on social media's impact on young persons' mental health. From a psychological perspective, the themes cover general anxiety and stress, self-esteem and body image concerns, hampering real-world social skills and community building. From a sociological perspective, themes emerge around the shaping of self-perception, the influences on social interactions, and mixed effects on mental health. From an educational studies perspective, themes focus on integrating digital literacy into curriculum development, addressing the psychosocial impact, adapting for special needs, leadership in the digital era, and developing policies to protect students' mental health, and social media as a societal mirror. From a political science perspective, themes focus on governance and policy, ethical implications, global mental health, economic influence, and cross-cultural impact of social media on young people. From a philosophy perspective, themes explore digital identity, information perception, ethics, logical reasoning, aesthetics, and governance. From a media studies perspective, themes focus on how social media shapes identity, socialisation, media literacy, fostering online communities, and affects mental health. From a linguistics perspective, the themes focus on social media distorts reality, affects self-esteem, leads to mental strain, fuels cyberbullying and peer pressure, and has both positive and negative influences. From a social work perspective, themes focus on hindering social skills, damages to self-esteem, enabling cyberbullying, misinformation, and the triggering of addictive behaviours impacting mental health. From an anthropology perspective, the themes focus on how digital culture shapes identity, affects development, changes communication, and creates a unique social structure within social media platforms. From a Health Science perspective, the themes focus on impacts to psychosocial aspects, influences on behaviour, effects on mental health, shaping of perception, and fostering of both positive and negative community connections.

Stage 2 of the analysis establishes superordinate themes by integrating the ten disciplinary perspectives separately for primary and secondary headteachers.

For primary headteachers (Table 1), five superordinate themes are identified. First, **Self-Perception and Identity Formation**, describing how social media shapes young people's self-image and identities, highlighting the gap between digital personas and real selves. It draws from several disciplines, including Sociology, Psychology, and Anthropology, emphasising the role of cultural norms and unrealistic beauty standards. Second, **Social Skills and Peer**

**Table 1. Superordinate themes from the thematic analysis of primary school headteacher narratives around social media effects on young people's mental health.**

| | Primary Headteachers | |
|---|---|---|
| **Superordinate Theme** | **Contributing Themes** | **Over-arching theory of Superordinate Theme** |
| **Self-Perception and Identity Formation** | **Sociological:** Social Comparison and Self-Perception. **Psychological:** Influence on Self-Perception and Identity Formation. **Education Studies:** Influence on Self-Perception and Identity. **Media Studies:** Influence of Media on Self-Perception and Identity Formation. **Linguistics:** Influence on Self-Perception and Body Image. **Social Work:** Influence on Self-Perception and Body Image. **Anthropology:** Cultural Norms and Expectations. **Health Sciences:** Social Comparison and Self-esteem | Interplay Between Digital Identity and Real Self. Social media significantly influences self-perception and identity formation, particularly in shaping cultural norms and expectations. This dynamic interplay between an individual's digital identity and real self can lead to discrepancies in self-image and self-worth, impacting mental health. |
| **Social Skills and Peer Interactions** | **Sociological:** Digital Peer Interaction and Cyberbullying. **Psychological:** Social Skills and Peer Interaction. **Education Studies:** Social and Emotional Learning, Cyberbullying and Online Safety. **Media Studies:** Impact of Digital Communication on Social Skills. **Linguistics:** Cyberbullying and Peer Pressure, Disruption of Real-World Interactions. **Social Work:** Online Bullying and Peer Pressure. **Anthropology:** Digital Interaction and Communication | Altered Social Interaction Dynamics. The digital landscape has transformed traditional social skills and peer interactions. The prevalence of cyberbullying and digital peer pressure, coupled with a reduction in face-to-face interactions, affects the development of healthy social skills and emotional well-being. |
| **Role in Mental Health and Well-being** | **Sociological:** Constant Connectivity and Mental Overload, Influence on Physical and Mental Health. **Psychological:** Exposure to Inappropriate Content and Cyberbullying. **Education Studies:** Cyberbullying and Online Safety. **Media Studies:** Psychological and Emotional Impact of Media Consumption. **Linguistics:** Impact on Mental Well-being and Anxiety. **Social Work:** Impact on Mental Health and Well-being, Addictive Nature and Disruption of Social Skills. **Health Sciences:** Digital Addiction and Mental Well-being, Cyberbullying and Peer Pressure | Complex Relationship with Mental Health. Social media's role in mental health is multifaceted, encompassing issues of constant connectivity, mental overload, exposure to inappropriate content, and cyberbullying. This complex relationship calls for nuanced understanding and interventions to support mental well-being. |
| **Digital Literacy and Critical Thinking** | **Education Studies:** Digital Literacy and Critical Thinking. **Media Studies:** Media Literacy and Digital Savvy. **Philosophy:** Perception and Reality, Cognitive and Emotional Development. **Health Sciences:** Parental Influence and Modeling | **Need for Enhanced Digital Literacy and Critical Thinking.** In the digital era, the ability to critically assess and engage with online content is essential. This includes developing digital literacy and critical thinking skills, which are crucial for navigating the challenges posed by social media. |
| **Governance, Policy, and Cultural Influence** | **Political Science:** Policy Impact on Social Behaviour, Globalised Cultural Influence, Governance and Regulation of Digital Spaces. **Philosophy:** Social Dynamics and Inclusion. **Education Studies:** Parental and Community Involvement. **Anthropology:** Social Connectivity and Isolation | **Impact of Governance and Cultural Contexts**: The influence of social media extends beyond individual usage, shaping broader societal norms and cultural influences. Governance and policy play a crucial role in regulating digital spaces and managing their impact on young people's mental health. |

**Interactions**, focused on altered social dynamics due to social media, this theme covers aspects like cyberbullying and the impact on face-to-face communication skills, integrating insights from Sociology, Education Studies, and Media Studies. Third, Direct **Mental Health and Well-being**, addressing the complex relationship between social media and mental health, this theme encompasses constant connectivity, mental overload, and exposure to inappropriate content, with contributions from disciplines like Health Sciences and Social Work. Fourth, **Digital Literacy and Critical Thinking**, highlighting the importance of developing digital literacy and critical thinking in the digital era. Contributions from Education Studies, Media Studies, and Philosophy underscore the need for enhanced skills to navigate digital challenges. Lastly, Governance**, Policy, and Cultural Influence**, focussing on the broader societal impact of social media, including the role of governance and policy in managing its influence on young people's mental health, drawing from Political Science and Anthropology.

For secondary headteachers (Table 2), the analysis reveals similar themes. First, **Identity and Self-Perception**, focussing on the profound influence of social media on identity construction and self-perception, mediated by cultural and societal norms and the portrayal of idealised images on digital platforms. Second, **Social Interaction and Communication Skills**, addressing the shift in socialisation dynamics, highlighting the transition from traditional to

**Table 2. Superordinate themes from the thematic analysis of secondary school headteacher narratives around social media effects on young people's mental health.**

| Superordinate Theme | Secondary Headteachers | |
|---|---|---|
| | Contributing Themes | Over-arching theory of Superordinate Theme |
| Identity and Self-Perception | **Sociological:** Influence on Self-Perception and Identity Formation. **Psychological:** Self-Esteem and Body Image Issues. **Media Studies:** Digital Influence on Identity Formation. **Linguistics:** Influence on Self-Image and Confidence. **Philosophy:** Digital Identity Formation. **Anthropology:** Digital Culture and Identity Formation. **Health Sciences:** Psychosocial Impact (related to self-esteem) | Focussed on identity and social norms. Social media profoundly influences the construction of identity and self-perception. This process is mediated by prevailing cultural and societal norms, which are often amplified and distorted in digital spaces. The continuous engagement with idealised and often unrealistic portrayals on social media platforms can lead to issues surrounding body image, self-esteem, and a general sense of inadequacy. |
| Social Interaction and Communication Skills | **Sociological:** Social Interaction and Community Building. **Psychological:** Social Skills and Real-World Interaction. **Education Studies:** Inclusive Digital Practices for Social, Emotional and Mental Health Needs Education. **Media Studies:** Virtual Socialisation and Communication Dynamics. **Linguistics:** Constant Connectivity and Mental Overload. **Social Work:** Social Skills and Interpersonal Relationships. **Anthropology:** Communication Patterns and Language Use | Shift in Socialisation Dynamics. The evolution of communication from face-to-face interactions to digital mediums has altered the fundamental nature of social interactions and relationships. This shift has implications for developing social skills, especially among younger populations, potentially leading to difficulties in real-world interactions and a sense of isolation, despite being more connected than ever. |
| Mental Health and Well-being | **Sociological:** Mental Health and Well-being. **Psychological:** Anxiety and Stress, Peer Influence and Cyberbullying. **Education Studies:** Psychosocial Impact of Social Media. **Political Science:** Globalisation of Mental Health Issues. **Media Studies:** Media-Induced Psychological Impacts. **Social Work:** Cyberbullying and Online Harassment, Addictive Behaviours and Mental Health. **Health Sciences:** Emotional Well-being and Mental Health | Complex Influence on Mental Health. The impact of social media on mental health is multifaceted, encompassing both positive and negative aspects. While it can be a source of stress, anxiety, and cyberbullying, it also offers opportunities for community building and support, particularly for marginalised groups. This dual nature necessitates a balanced approach in addressing mental health concerns related to social media use. |
| Digital Literacy and Information Perception | **Education Studies:** Digital Literacy and Curriculum Development. **Philosophy:** Information Perception and Influence. **Media Studies:** Media Literacy and Critical Consumption. **Linguistics:** Perceived Reality vs. Actual Reality. **Health Sciences:** Information and Perception Management | Necessity for Enhanced Digital Literacy. The ability to critically engage with and interpret digital content is crucial in the modern digital era. This requires integrating digital literacy into educational curricula and broader societal initiatives, equipping young individuals with the skills to navigate the complexities of digital information and its impact on their perceptions and beliefs. |
| Governance and Policy in Digital Spaces | **Political Science:** Digital Governance and Policy, Social Media as a Socio-Political Tool. **Education Studies:** Leadership in Digital Era, Policy Development for Digital Well-being. **Philosophy:** Governance and Social Structures Online. **Anthropology:** Social Media as a Social Structure | Role of Governance and Policy in Digital Spaces. Effective governance and policy-making are essential to mitigate the risks and maximise the benefits of social media use. This involves not just the regulation of content and platforms but also education and awareness campaigns, support systems, and international cooperation to address the global nature of digital influences. |

digital communication methods and their implications for developing social skills and real-world interactions. Third, **Influence on Well-being**, discussing the multifaceted impact of social media on mental health, including both stress and anxiety-inducing factors and opportunities for community support, especially for marginalised groups. Fourth, **Digital Literacy and Information Perception**, focussing on the necessity of digital literacy in modern education, emphasising the critical engagement with digital content and its impact on perceptions and beliefs. Lastly, **Governance and Policy in Digital Spaces**, exploring the crucial role of governance and policy in regulating digital spaces, focusing on content regulation, education, awareness, and international cooperation.

## Discussion

The study, through its innovative methodology employing ChatGPT-4 for thematic analysis, provides an unprecedented multi-disciplinary perspective on the impact of social media on young peoples' mental health. The sample of 492 UK school headteachers, including those with roles in headship, deputy headship, and pupil welfare, offered a rich and varied dataset for analysis, though they tend to reflect a predominantly negative perspective on the impact of social media on young people's mental health, highlighting concerns over issues such as

depression, anxiety, and self-esteem, self-perception, health and body image, social skills, cyber-bullying, digital literacy, governance, and cultural norms. The results, separated by primary and secondary education levels, reveal significant insights into the multifaceted nature of social media's influence on mental health across different developmental stages. This provides a comprehensive examination of the effects of social media on young peoples' mental health through the lens of UK school headteachers.

Drawing from a multidisciplinary approach across primary and secondary sectors that includes psychology, media studies, sociology, anthropology, linguistics, social work, philosophy, education, and health sciences, the study highlights both the potential negative and positive aspects of social media use. The majority of assessments from the headteachers suggest a more negative view on the effects of social media on mental health, which is consistent with the findings of the literature [65]. On the negative side, headteachers noted concerns about social media exacerbating issues of anxiety, depression, cyberbullying, and sleep disturbances. The constant connectivity and exposure to unrealistic standards on social media platforms were seen as contributing to negative self-perception, identity formation issues, and increased mental health challenges among young people. These insights underscore the complex relationship between social media use and mental well-being, highlighting the need for tailored interventions and policies to mitigate these adverse effects. Conversely, the study also identifies positive aspects of social media, such as fostering social connectedness, providing support networks, and facilitating identity exploration. The headteachers acknowledged the role of social media in offering a space for young people to find communities, engage in social activism, and express themselves creatively. This dual perspective emphasises the refined impact of social media, suggesting that while there are significant concerns, there are also opportunities for positive engagement that can support mental health.

In terms of the themes for developing a multi-disciplinary understanding of social media effects on mental health, five similar themes emerged for primary and secondary headteachers. For primary headteachers, five superordinate themes emerged: Self-Perception and Identity Formation, Social Skills and Peer Interactions, Direct Mental Health and Well-being, Digital Literacy and Critical Thinking, and Governance, Policy, and Cultural Influence. These themes highlight the complex interplay between social media and various aspects of young people's lives, including identity formation, social interaction, and mental well-being. The synthesis of perspectives from diverse disciplines such as Sociology, Psychology, Education Studies, and Anthropology highlights the nuanced ways in which social media impacts young children. Additionally, the secondary headteachers' responses centred around similar yet distinctly developed themes like Identity and Self-Perception, Social Interaction and Communication Skills, Influence on Well-being, Digital Literacy and Information Perception, and Governance and Policy in Digital Spaces. This reflects a more mature understanding of social media's role in shaping adolescents' identities and social interactions, as well as the importance of digital literacy and policy in navigating these digital spaces.

The analysis of responses from both primary and secondary school headteachers revealed common themes, leading to the development of a new model, we name as the Comprehensive Digital Influence Model (See Table 3), focusing on five key areas:

- Self-Identity and Perception Formation: Both primary and secondary levels emphasise the significant role of social media in shaping self-image, identity, and self-esteem. This convergence indicates a universal effect of social media in moulding young people's self-conception across all ages [66–70].

- Social Interaction Skills and Peer Communication: A shared concern is the impact of social media on social skills and peer interactions, including the prevalence of cyberbullying. This

**Table 3. The comprehensive digital influence model, indicating different emphases between primary and secondary school contexts.**

| Primary Schools | Comprehensive Digital Influence Factors | Secondary Schools |
|---|---|---|
| Social Media's impact on children's early social skills, identity formation | **Self-Identity and Perception Formation** | Nuanced view of older children's psychological development in digital society. |
| Developing basic social skills, addressing digital communication's impact, cyberbullying focus | **Social Interaction Skills and Peer Communication** | Complex social dynamics, real-world skills, digital impact on mental health. |
| Social media's link to mental health, overload, content exposure | **Indicators of Mental and Emotional Well-Being** | Layered view on social media's varied impacts on adolescent health. |
| Fostering media literacy in children by discerning reality from digital. | **Digital Literacy, Critical Thinking, and Information Perception** | Advanced digital literacy through curriculum integration, sophisticated content engagement |
| Foundation approach: policy impact, global culture, enhanced parental involvement. | **Governance, Policy, and Cultural Influence in Digital Spaces** | Sophisticated themes in digital governance, policy, educational leadership's role |

highlights a widespread apprehension about the transformation of traditional social dynamics due to digital interactions, evident in both educational stages [71–73].

- Mental and Emotional Well-Being: The analyses from both primary and secondary levels acknowledge the complex relationship between social media use and mental health issues like anxiety and stress. This mutual focus underscores the recognition of social media as a pivotal factor in young people's mental health, with both its positive and negative aspects [20, 74].

- Digital Literacy, Critical Thinking, and Information Perception: There is an emphasis on the necessity for enhanced digital literacy and critical thinking skills. This reflects a common understanding of the importance of empowering young people with the skills to navigate and critically evaluate digital content [75–78].

- Governance, Policy, and Cultural Influence in Digital Spaces: Both educational levels underscore the importance of governance and policy in managing digital spaces and their influence on mental health. This convergence suggests a consensus on the need for structured regulation and guidance in the digital realm, highlighting the crucial role of governance and societal influence in mitigating the challenges posed by digital media [79–81].

Within the Comprehensive Digital Influence Model (also see Table 3), we are also able to note that in terms of age-related psychological development, the thematic focus of primary and secondary schools diverges notably. Primary school themes are predominantly centred on foundational aspects, encompassing the formation of basic social skills, children's initial interactions with digital media, and the early impact of social media on self-perception and identity formation. In contrast, secondary school themes explore more intricate psychological and social dynamics. For example, in terms of the first dimension, Self-Identity and Perception Formation, in primary schools the focus is on the basic development of social skills and the initial stages of identity formation. Concerns are predominantly centred on how social media influences the early shaping of self-image and identity in children [82]. In secondary schools, a more nuanced view is presented, delving into the interactions between psychological factors, social dynamics, digital literacy, and societal structures. This approach reflects a deeper understanding of older children who are navigating more complex stages of psychological development [83]. Second is Social Interaction Skills and Peer Communication. In primary schools, the concentration is on developing basic social skills and mitigating the immediate effects of digital communication, with a significant focus on the immediate impact of cyberbullying [84,

85]. However, in secondary schools, the focus is more on complex social dynamics. This encompasses the development of skills needed for real-world interactions, understanding the role of digital platforms in social interaction, and grappling with the impact of constant connectivity on mental health [32]. Third is: Mental and Emotional Well-Being. In primary schools, the emphasis lies on the direct relationship between social media and mental health. This includes addressing issues like constant connectivity, mental overload, and children's initial exposure to potentially inappropriate content [86]. However, in secondary schools, Headteachers offer a more layered perspective of mental health, acknowledging both the positive and negative aspects of social media use. This stance indicates a more profound comprehension of the diverse impacts on adolescent mental health [34]. Fourth is Digital Literacy, Critical Thinking, and Information Perception. In primary schools, the focus is on rudimentary stages of developing these skills. Efforts are made to help children discern the difference between reality and digital portrayals, thereby laying the groundwork for media literacy [87]. Among secondary schools, the emphasis is on the advanced development of digital literacy. This includes integrating these concepts into the curriculum, and fostering sophisticated information perception and management skills, indicating a deeper engagement with digital content [88, 89]. Fifth is Governance, Policy, and Cultural Influence in Digital Spaces. Among primary schools, a foundational approach is suggested, focusing on the impact of policy, global cultural influences, and the need for enhanced positive parental and community involvement [90]. Among secondary schools, headteacher discussions revolve around more sophisticated themes, such as digital governance, the development of policies for digital well-being, and the role of educational leadership in navigating the challenges of the digital era [91, 92].

The study delivers significant insights across various domains, notably by establishing a more holistic, nuanced understanding of the complex interplay between social media and young people's mental health, and by formulating succinct, policy and educationally relevant recommendations. Employing ChatGPT-4 for thematic analysis, our methodology introduces a comprehensive multidisciplinary lens to examine the complex interactions between social media use and the mental health of young individuals. This innovative approach not only enhances the granularity of our findings but also establishes a new precedent for future research methodologies within this sphere. The introduction of the Comprehensive Digital Influence Model—based on the study's findings—provides a structured, holistic framework that encapsulates the multifaceted impacts of social media across different developmental stages and thematic areas. By synthesizing perspectives from diverse disciplines such as sociology, psychology, education, and anthropology, the model offers a more holistic and complex understanding of social media's influence than is typically achieved through single-discipline studies. This comprehensive approach is especially valuable for mental health professionals and educators, equipping them with a nuanced understanding that can inform therapeutic interventions and educational curricula designed to mitigate the risks associated with social media use, such as cyberbullying and addiction. Regarding policy implications, our findings are instrumental for policymakers in shaping educational content that addresses critical areas like digital literacy, mental health, and social skills, tailored to cater to various developmental needs. The study highlights the importance of creating informed governance and policies that effectively manage and regulate digital media's influence on youth, advocating for partnerships with technology companies to establish safer online environments. It also suggests enhancing parental and community involvement to better manage children's digital interactions, emphasizing a collaborative approach to digital education and safety. Furthermore, the insights derived from this research are crucial for the development of educational practices that foster not only digital literacy but also critical thinking and social-emotional learning. By integrating these elements into school curricula, the study advocates for a more informed and proactive

approach to social media use among young people. This includes empowering students to critically assess online content and engage in positive online interactions, thereby equipping them with the tools needed to navigate the complexities of digital spaces effectively. In essence, this study offers valuable contributions to theoretical frameworks, policy development, and educational practices, underscoring the need for an integrated approach that combines insights from multiple disciplines to fully address the intricate dynamics of social media's impact on young people's mental health. This makes it a pivotal resource for stakeholders ranging from academic researchers to policymakers and educators, all of whom play a critical role in shaping the digital landscape to support the well-being of young users.

In examining the study, it is pivotal to acknowledge the unique expertise of the participants —UK school headteachers—whose insights form the foundation of this important work. Their expert views offer a distinctive perspective on the impact of social media on young peoples' mental health, making this study a substantial contribution to the field. However, while the study stands as a foundational piece of work, leveraging the specialised knowledge of headteachers, it also encounters certain limitations. The sample, although rich with the professional insights of headteachers, may not encompass the full spectrum of global perspectives, particularly those of students, parents, and other educators, potentially affecting the universality of the findings across various cultural, socioeconomic, or educational contexts. Methodologically, while the use of ChatGPT-4 for thematic analysis is innovative, it might have intrinsic limitations in fully grasping the qualitative nuances that human researchers, particularly those with a deep understanding of educational contexts, could interpret. Moreover, it's important to consider the political biases noted in ChatGPT-4 [45–47], as these may influence how it processes and interprets data. While the influence of such biases is expected to be limited in the context of analyzing educational impacts of social media on mental health among young people, they are still worth careful consideration. Also, the single question, though effective in gaining a large number of views and allows respondents to focus on a single issue, limits the ability to understand causal relationships or the enduring effects of social media on young peoples' mental health. Future research, building on this foundation laid by experts in education, could explore several avenues. Building on this research to build on a wider array of perspectives, such as those from students and parents, would provide a more comprehensive view of the issue–but doing so within the proposed model may be necessary to provide context or development of research with those views. As such there are three possible ways in which this can be done to refine and develop the model. First, integrating developmental psychology more explicitly could provide a deeper alignment of the model with the cognitive, emotional, and social development stages of children and adolescents. This approach would benefit from the inclusion of insights from developmental psychologists to refine how social media impacts are framed within different age groups. Second, expanding data sources to include direct input from children and adolescents through qualitative studies or surveys could enrich the model by incorporating firsthand perspectives on social media's influence, adding depth and accuracy. Lastly, refining the model parameters to account for variables such as socio-economic status, cultural background, and individual psychological traits would enable the creation of more nuanced interventions and policies, tailored to meet the diverse needs of different student populations, thereby enhancing the model's utility and applicability.

This study, leveraging ChatGPT-4's thematic analysis and insights from 492 UK school headteachers, presents the Comprehensive Digital Influence Model (CDIM), a novel framework encapsulating the complex impact of social media on young peoples' mental health. The CDIM, distilled from both primary and secondary school perspectives, highlights key areas: Self-Identity and Perception Formation, Social Interaction Skills, Mental and Emotional Well-Being, Digital Literacy and Critical Thinking, and Governance in Digital Spaces. This study

not only advances methodological approaches in educational and psychological research but also offers invaluable insights for policy development and educational practices. It emphasises the urgent need for structured digital literacy programs, nuanced mental health strategies, and effective governance in digital media, particularly tailored for varying developmental stages. As a pioneering exploration into the digital influences shaping young minds, this work sets a precedent for future interdisciplinary research and policymaking, marking the crucial role of comprehensive, age-appropriate strategies in navigating the digital era's challenges and opportunities.

## Supporting information

**S1 Table. Instructions for multidisciplinary analysis using ChatGPT-4 (Used separately for each discipline).**
(DOCX)

**S2 Table. Disciplinary and sub-disciplinary frameworks for thematic analysis with a rationale for inclusion.**
(DOCX)

**S3 Table. Instructions for multidisciplinary analysis using ChatGPT-4 (Separately for primary and secondary schools).**
(DOCX)

**S4 Table. Thematic analysis of primary school headteacher narratives around social media effects on young people's mental health.**
(DOCX)

**S5 Table. Thematic analysis of secondary school headteacher narratives around social media effects on young people's mental health.**
(DOCX)

## Author Contributions

**Conceptualization:** John Maltby, Sanjiv Nichani.

**Data curation:** John Maltby, Thooba Rayes, Sulaimaan Sharif, Maryama Omar, Sanjiv Nichani.

**Formal analysis:** John Maltby, Antara Nage.

**Investigation:** John Maltby, Thooba Rayes.

**Methodology:** John Maltby, Sanjiv Nichani.

**Project administration:** John Maltby.

**Writing – original draft:** John Maltby.

**Writing – review & editing:** John Maltby, Thooba Rayes, Antara Nage, Sulaimaan Sharif, Maryama Omar, Sanjiv Nichani.

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
