## [Decision Letter · Decision Letter 0]

7 May 2024

PONE-D-24-07558Synthesising perspectives: Crafting an Interdisciplinary view of social media's impact on young people's mental health.PLOS ONE

Dear Dr. Maltby,

Thank you for submitting your manuscript to PLOS ONE. After careful consideration, we feel that it has merit but does not fully meet PLOS ONE’s publication criteria as it currently stands. Therefore, we invite you to submit a revised version of the manuscript that addresses the points raised during the review process.

We look forward to receiving your revised manuscript.

Kind regards,

Luis Hernan Contreras Pinochet, Ph.D.

Academic Editor

PLOS ONE

Journal Requirements:

4. Please include a caption for figure 1.

5. Please upload a copy of Figure 1, to which you refer in your text on page 23. If the figure is no longer to be included as part of the submission please remove all reference to it within the text.

**Additional Editor Comments:**

Dear Authors,

Thank you for submitting the article entitled "Synthesising Perspectives: Crafting an Interdisciplinary View of Social Media's Impact on Young People's Mental Health" for review in our magazine. After a detailed analysis by reviewers specialized in the field, it became evident that the work holds significant potential but also identified specific areas requiring improvement to maximize its impact.

The reviewers primarily highlighted the need for enhancements in the article's theoretical and methodological aspects. Their observations provide valuable insights to fortify the study's theoretical foundation and methodological robustness.

I would like to request that all adjustments and requests highlighted by the reviewers be addressed in the new version of the article. This encompasses, but is not confined to:

Revising and deepening the theoretical framework, ensuring the consideration and coherent integration of all pertinent approaches.

Enhancing the methodology, which entails clarifying procedures, justifying methodological choices, and adequately addressing potential limitations.

Integrating any additional suggestions or comments provided by the reviewers to enhance the overall quality of the article.

I underscore the importance of these revisions to ensure that the article fulfills its maximum potential and merits consideration for publication in our journal.

Please do not hesitate to reach out should you have any questions or require further clarification regarding the reviewers' feedback. I am at your disposal to assist you throughout the revision process.

Thank you in advance for your dedication to improving the article. I eagerly anticipate the revised version.

Yours sincerely,

Reviewers' comments:

Reviewer's Responses to Questions

**Comments to the Author**

1. Is the manuscript technically sound, and do the data support the conclusions?

Reviewer #1: Yes

Reviewer #2: Yes

2. Has the statistical analysis been performed appropriately and rigorously? 

Reviewer #1: N/A

Reviewer #2: N/A

3. Have the authors made all data underlying the findings in their manuscript fully available?

Reviewer #1: Yes

Reviewer #2: Yes

4. Is the manuscript presented in an intelligible fashion and written in standard English?

Reviewer #1: Yes

Reviewer #2: Yes

5. Review Comments to the Author

Reviewer #1: Regarding the Comprehensive Digital Influence Model, how robust are the identified key themes in capturing the complex dynamics between social media and mental health among young people? Are there any potential limitations or areas where further refinement may be necessary?

In terms of practical implications, how do the insights provided by the study inform the actions of educators, policymakers, and mental health professionals? Are there specific recommendations or strategies proposed that are actionable and feasible in addressing the challenges posed by social media in supporting the mental well-being of young individuals?

Reviewer #2: The paper "Synthesising perspectives: Crafting an Interdisciplinary view of social media's impact on young people's mental health" explores a highly relevant theme for practice as well as for various fields of research.

The paper stands out for the clarity of the written text, rigor in the application of the research method, use of an interdisciplinary approach (combining various knowledge perspectives), and discussion of the practical implications of the study, aiming to demonstrate how the research findings can impact the formulation of public policies.

In essence, the paper’s main contribution relies on the development of a method to perform thematic analysis using ChatGPT, which could be replicated in other qualitative research studies.

The following remarks seek to summarize the key areas where the paper needs improvement:

1) Include a Research Question (RQ): The first chapter (Introduction) is effective in showing the topic and clearly stating the research gap, highlighting the lack of studies exploring the effect of social media on young people's mental health from an interdisciplinary perspective. However, it would be advisable to include a research question at the end of this chapter, aiming to state the research scope.

2) Methods: Page 8 contains the following excerpt: “(...) ChatGPT offers a consistent, unbiased processing capability, crucial for handling the subtle and often subjective nature of qualitative data (...).” Considering the use of ChatGPT is the core of the paper’s research design, there is a lack of references/citations that support the author's choice in the use of this tool. In that excerpt, it was mentioned the “unbiased processing capability” of ChatGPT; certainly, that is a questionable statement that demands a more in-depth discussion to justify the use of ChatGPT.

Here are some examples of studies that doubt the “unbiased processing capability” of ChatGPT:

[1] Rozado, D. (2023). The political biases of chatgpt. Social Sciences, 12(3), 148.

[2] Ray, P. P. (2023). ChatGPT: A comprehensive review on background, applications, key challenges, bias, ethics, limitations and future scope. Internet of Things and Cyber-Physical Systems.

[3] Motoki, F., Pinho Neto, V., & Rodrigues, V. (2024). More human than human: Measuring ChatGPT political bias. Public Choice, 198(1), 3-23.

3) Results are essentially descriptive, and there is a lack of discussion of findings with the literature: The main weakness of the paper relies on the descriptive aspect of the chapters Results and Discussion. In essence, these chapters describe the outputs from the thematic analysis without any analysis or discussion with the previous literature on the impact of social media on young people's mental health (from a disciplinary or interdisciplinary perspective) (there is no citation from pages 12 to 26).

4) Explain how the interdisciplinary approach adopted in the paper differs from other studies: The authors mention that the differential of the interdisciplinary approach carried out in the paper is that it allows a more precise analysis of this complex phenomenon, overcoming the limitations of a specialized analysis considering only one field of study. However, the paper lacks a more detailed discussion to demonstrate what would be the differential of the findings using this interdisciplinary approach? How are the results presented similar to (or different from) other studies analyzing the impact of social media on young people's mental health?

6. PLOS authors have the option to publish the peer review history of their article (what does this mean?). If published, this will include your full peer review and any attached files.

Reviewer #1: No

Reviewer #2: No

---

## [Author Response · Author response to Decision Letter 0]

8 Jun 2024

RESPONSE TO REVIEWERS

Thank you for considering the recent submission and for your encouraging response regarding the revision and resubmission of the manuscript. I extend my gratitude to the reviewers for their insightful, helpful, patient, and constructive feedback. In response to the reviewers' comments, I have thoroughly reviewed the paper and made numerous significant changes and additions. I have also corrected some typing errors and I apologise for the presence of those. To provide clarity the revised text corresponding to each response is included alongside the response with the location of the change in the revised manuscript also presented.

General Comments

1. Please include your full ethics statement in the ‘Methods’ section of your manuscript file. In your statement, please include the full name of the IRB or ethics committee who approved or waived your study, as well as whether or not you obtained informed written or verbal consent. If consent was waived for your study, please include this information in your statement as well.

>>>>>This is now provided on Page 7, Lines 1 to 7 of the revised manuscript as follows: 

“Ethics approval for the study was granted by the University of Leicester Psychology Ethics Board, with the reference number 42292-jm148-ls&visionsciences,schof. In alignment with ethical guidelines, informed consent was obtained from all participants. This consent was facilitated through an online survey, where respondents indicated their consent by affirmatively selecting the consent option on the digital form confirming consent statements. This consent option confirmed that participants were fully informed and voluntarily agreed to participate in the study. “ 

2. Please include a caption for figure 1.

>>>>>> Apologies there is no Figure 1. This is an error – and is Table 3. Any reference has been removed. 

3. Please upload a copy of Figure 1, to which you refer in your text on page 23. If the figure is no longer to be included as part of the submission please remove all reference to it within the text.

>>>>>> Apologies there is no Figure 1. This is an error – and is Table 3. Any reference has been removed.

>>>>> We have reviewed the reference list, double-checked it for errors, and updated it. Additionally, in response to Reviewer 2's concern (point 3) about the lack of literature, we have revised the manuscript to include over 50 more references.

Editor comments

The reviewers primarily highlighted the need for enhancements in the article's theoretical and methodological aspects. Their observations provide valuable insights to fortify the study's theoretical foundation and methodological robustness.

I would like to request that all adjustments and requests highlighted by the reviewers be addressed in the new version of the article. This encompasses, but is not confined to:

1. Revising and deepening the theoretical framework, ensuring the consideration and coherent integration of all pertinent approaches.

>>>>> Our theoretical framework now emphasizes the integration of multiple disciplines to develop a more holistic understanding of social media's impact on mental health, recognizing the complexity of these interactions. The literature review has been significantly expanded to include over 50 new references, providing a robust theoretical foundation. Theoretical insights have been translated into practical, policy-relevant recommendations. We provide specific strategies for integrating digital literacy into curricula, guidelines for mental health interventions, and policy recommendations to address the challenges posed by social media. We have added a detailed discussion of the strengths and limitations of the Comprehensive Digital Influence Model. This section highlights the robustness of the model in capturing the complex dynamics between social media and mental health among young people.

By revising and deepening the theoretical framework through these comprehensive and interdisciplinary approaches, we ensure a more nuanced and holistic understanding of social media's impact on young people's mental health, thereby enhancing the persuasiveness of the article.

2. Enhancing the methodology, which entails clarifying procedures, justifying methodological choices, and adequately addressing potential limitations.

>>>>> The methodology has been clarified and enhanced by providing a more in-depth justification for using ChatGPT-4, including references to relevant studies that support its application in qualitative research. We addressed concerns about its unbiased processing capability by discussing its limitations and citing studies that highlight potential biases. This includes a discussion of political biases in ChatGPT-4 and their minimal expected influence in this context.

3. Integrating any additional suggestions or comments provided by the reviewers to enhance the overall quality of the article.

As outlined below we have fully considered these comments and are detailed below, but summarised here as follows:

Reviewer #1:

• Key Themes Robustness: We have added a discussion on Page 33, Lines 1-12 about the robustness of the Comprehensive Digital Influence Model and potential areas for refinement, including the integration of developmental psychology, direct input from children and adolescents, and accounting for socio-economic and cultural variables.

• Practical Implications: The practical implications section has been expanded on Page 30, Line 15 to Page 32, Line 2, providing specific, actionable recommendations for educators, policymakers, and mental health professionals.

Reviewer #2:

• Research Question: A research question has been added at the end of the Introduction on Page 6, Lines 16-24 to clearly define the scope of our research.

• Methodological References: We provided references and a more in-depth discussion justifying the use of ChatGPT-4, addressing its potential biases, on Page 9, Line 22 to Page 10, Line 25.

• Results and Discussion: We expanded the Results and Discussion sections to include over 50 new references, developing a critical analysis of the literature on Pages 12-13 and Pages 26-27, and comparing primary and secondary school analyses on Pages 29-30.

• Interdisciplinary Approach: The value of the interdisciplinary approach is highlighted in the Introduction and Discussion sections on Pages 5-6 and Pages 30-32, demonstrating how it provides a more holistic and nuanced understanding compared to single-discipline studies.

4. I underscore the importance of these revisions to ensure that the article fulfills its maximum potential and merits consideration for publication in our journal.

>>>>> We have meticulously addressed each reviewer's suggestions, significantly enhancing the theoretical framework, methodology, and integration of interdisciplinary perspectives. We think these revisions ensure that the article presents a comprehensive, nuanced analysis of the complex interplay between social media and young people's mental health, making it a valuable resource for stakeholders and worthy of consideration for publication.

Reviewer #1:

1. Regarding the Comprehensive Digital Influence Model, how robust are the identified key themes in capturing the complex dynamics between social media and mental health among young people? Are there any potential limitations or areas where further refinement may be necessary?

>>>> We acknowledge the need to further elaborate on the robustness of the identified key themes in capturing the complex dynamics between social media and mental health among young people. To address this, we have added a section discussing the strengths and limitations of the Comprehensive Digital Influence Model, highlighting areas for potential refinement and future research. The revised section is on Page 33, Lines 1 to 12, as follows:

“As such there are three possible ways in which this can be done to refine and develop the model. First, integrating developmental psychology more explicitly could provide a deeper alignment of the model with the cognitive, emotional, and social development stages of children and adolescents. This approach would benefit from the inclusion of insights from developmental psychologists to refine how social media impacts are framed within different age groups. Second, expanding data sources to include direct input from children and adolescents through qualitative studies or surveys could enrich the model by incorporating firsthand perspectives on social media's influence, adding depth and accuracy. Lastly, refining the model parameters to account for variables such as socio-economic status, cultural background, and individual psychological traits would enable the creation of more nuanced interventions and policies, tailored to meet the diverse needs of different student populations, thereby enhancing the model’s utility and applicability.”

2. In terms of practical implications, how do the insights provided by the study inform the actions of educators, policymakers, and mental health professionals? Are there specific recommendations or strategies proposed that are actionable and feasible in addressing the challenges posed by social media in supporting the mental well-being of young individuals? 

>>>>> The discussion on practical implications has been expanded to provide specific, actionable recommendations for educators, policymakers, and mental health professionals. We outline strategies for integrating digital literacy into curricula, guidelines for mental health interventions, and policy recommendations to address the challenges posed by social media. The revised section is on Page 30, Line 15 to Page 32, Line 2, as follows:

“The study delivers significant insights across various domains, notably by establishing a more holistic, nuanced understanding of the complex interplay between social media and young people's mental health, and by formulating succinct, policy and educationally relevant recommendations. Employing ChatGPT-4 for thematic analysis, our methodology introduces a comprehensive multidisciplinary lens to examine the complex interactions between social media use and the mental health of young individuals. This innovative approach not only enhances the granularity of our findings but also establishes a new precedent for future research methodologies within this sphere. The introduction of the Comprehensive Digital Influence Model—based on the study’s findings—provides a structured, holistic framework that encapsulates the multifaceted impacts of social media across different developmental stages and thematic areas. By synthesizing perspectives from diverse disciplines such as sociology, psychology, education, and anthropology, the model offers a more holistic and complex understanding of social media's influence than is typically achieved through single-discipline studies. This comprehensive approach is especially valuable for mental health professionals and educators, equipping them with a nuanced understanding that can inform therapeutic interventions and educational curricula designed to mitigate the risks associated with social media use, such as cyberbullying and addiction. Regarding policy implications, our findings are instrumental for policymakers in shaping educational content that addresses critical areas like digital literacy, mental health, and social skills, tailored to cater to various developmental needs. The study highlights the importance of creating informed governance and policies that effectively manage and regulate digital media's influence on youth, advocating for partnerships with technology companies to establish safer online environments. It also suggests enhancing parental and community involvement to better manage children’s digital interactions, emphasizing a collaborative approach to digital education and safety. Furthermore, the insights derived from this research are crucial for the development of educational practices that foster not only digital literacy but also critical thinking and social-emotional learning. By integrating these elements into school curricula, the study advocates for a more informed and proactive approach to social media use among young people. This includes empowering students to critically assess online content and engage in positive online interactions, thereby equipping them with the tools needed to navigate the complexities of digital spaces effectively. In essence, this study offers valuable contributions to theoretical frameworks, policy development, and educational practices, underscoring the need for an integrated approach that combines insights from multiple disciplines to fully address the intricate dynamics of social media’s impact on young people’s mental health. This makes it a pivotal resource for stakeholders ranging from academic researchers to policymakers and educators, all of whom play a critical role in shaping the digital landscape to support the well-being of young users.”

Reviewer #2:

1. Include a Research Question (RQ): The first chapter (Introduction) is effective in showing the topic and clearly stating the research gap, highlighting the lack of studies exploring the effect of social media on young people's mental health from an interdisciplinary perspective. However, it would be advisable to include a research question at the end of this chapter, aiming to state the research scope.

>>>>> A research question has been added at the end of the Introduction to clearly define the scope of our research. The revised section is on Page 6, Lines 16 to 24, as follows:

“This study aims to delve into the effects of social media on young people's mental health by merging insights from various academic fields. By synthesizing knowledge from disciplines such as psychology, sociology, education, and media studies, we form a holistic understanding of how social media influences the well-being of young individuals. Specifically, this study seeks to answer the research question: 'How do interdisciplinary perspectives enhance our understanding of the impact of social media on the mental health of young people?' This question aims to clarify the scope of our research and guide the investigation into the complex dynamics of social media's influence across different disciplinary lenses.”

2. Methods: Page 8 contains the following excerpt: “(...) ChatGPT offers a consistent, unbiased processing capability, crucial for handling the subtle and often subjective nature of qualitative data (...).” Considering the use of ChatGPT is the core of the paper’s research design, there is a lack of references/citations that support the author's choice in the use of this tool. In that excerpt, it was mentioned the “unbiased processing capability” of ChatGPT; certainly, that is a questionable statement that demands a more in-depth discussion to justify the use of ChatGPT.

Here are some examples of studies that doubt the “unbiased processing capability” of ChatGPT:

[1] Rozado, D. (2023). The political biases of chatgpt. Social Sciences, 12(3), 148.

[2] Ray, P. P. (2023). ChatGPT: A comprehensive review on background, applications, key challenges, bias, ethics, limitations and future scope. Internet of Things and Cyber-Physical Systems.

[3] Motoki, F., Pinho Neto, V., & Rodrigues, V. (2024). More human than human: Measuring ChatGPT political bias. Public Choice, 198(1), 3-23.

>>>>> Thank you for the details and references. We have provided a more in-depth justification for using ChatGPT-4, including references to relevant studies that support its application in qualitative research. Additionally, we address concerns about its unbiased p

---

## [Editor Report · Decision Letter 1]

2 Jul 2024

Synthesizing perspectives: Crafting an Interdisciplinary view of social media's impact on young people's mental health.

PONE-D-24-07558R1

Dear Dr. Maltby,

We’re pleased to inform you that your manuscript has been judged scientifically suitable for publication and will be formally accepted for publication once it meets all outstanding technical requirements.

Kind regards,

Luis Hernan Contreras Pinochet, Ph.D.

Academic Editor

PLOS ONE

Additional Editor Comments (optional):

Dear Authors,

Thank you for your detailed and thoughtful response regarding the revision and resubmission of your manuscript.

I am pleased to inform you that all the reviewers' queries and requests have been brilliantly addressed. The manuscript has considerably improved following the adjustments made during this latest revision.

We appreciate your efforts in ensuring the clarity and quality of the manuscript. We look forward to progressing with the publication process.

Best regards,
---

## [Editor Report · Acceptance letter]

5 Jul 2024

PONE-D-24-07558R1 

PLOS ONE

Dear Dr. Maltby, 

I'm pleased to inform you that your manuscript has been deemed suitable for publication in PLOS ONE. Congratulations! Your manuscript is now being handed over to our production team.

Kind regards, 

on behalf of

Dr. Luis Hernan Contreras Pinochet 

Academic Editor

PLOS ONE